# RenderMe-360: A Large Digital Asset Library and Benchmarks Towards High-fidelity Head Avatars

**Dongwei Pan**[1] , **Long Zhuo**[1*] , **Jingtan Piao**[2,5*] , **Huiwen Luo**[1*] , **Wei Cheng**[1*] , **Yuxin Wang**[1,3*] ,

**Siming Fan**[2] , **Shengqi Liu**[2] , **Lei Yang**[2] , **Bo Dai**[1] , **Ziwei Liu**[4] , **Chen Change Loy**[4] , **Chen Qian**[1] ,

**Wayne Wu**[1] , **Dahua Lin**[1,5] , **Kwan-Yee Lin**[1,5]

[1] Shanghai AI Laboratory     [2] SenseTime Research     [3] HKUST     [4] S-Lab, NTU
[5] CUHK
linjunyi9335@gmail.com

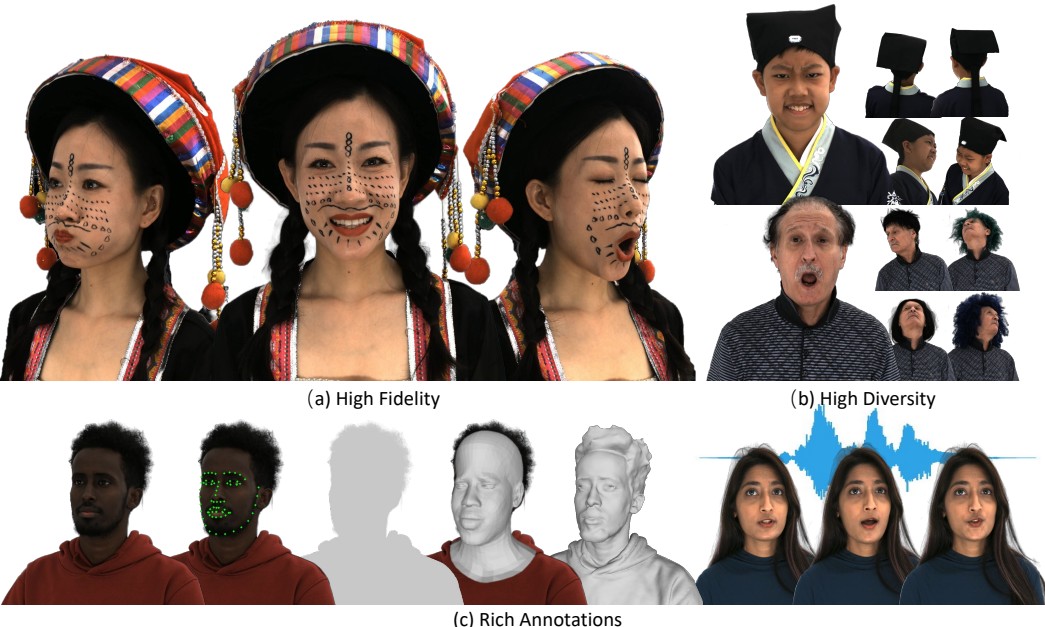

(a) High Fidelity                                 （b) High Diversity

(c) Rich Annotations

Figure 1: **Overview of RenderMe-360's core features.** We present a large digital asset library RenderMe-360 to facilitate the development of advanced research on high-fidelity head avatar synthesis. It has the characteristics of (a) high fidelity and (b) high diversity. Also, our dataset comes with (c) rich annotations.

## Abstract

Synthesizing high-fidelity head avatars is a central problem for computer vision and graphics. While head avatar synthesis algorithms have advanced rapidly, the best ones still face great obstacles in real-world scenarios. One of the vital causes is the inadequate datasets – 1) current public datasets can only support researchers to explore high-fidelity head avatars in one or two task directions; 2) these datasets usually contain digital head assets with limited data volume,

---

*Joint-second authors.

37th Conference on Neural Information Processing Systems (NeurIPS 2023) Track on Datasets and Benchmarks.

and narrow distribution over different attributes, such as expressions, ages, and accessories. In this paper, we present **RenderMe-360**, a comprehensive 4D human head dataset to drive advance in head avatar algorithms across different scenarios. It contains massive data assets, with $243+$ million complete head frames and over $800k$ video sequences from 500 different identities captured by multi-view cameras at 30 FPS. It is a large-scale digital library for head avatars with three key attributes: 1) High Fidelity: all subjects are captured in 360 degrees via 60 synchronized, high-resolution $2K$ cameras. 2) High Diversity: The collected subjects vary from different ages, eras, ethnicities, and cultures, providing abundant materials with distinctive styles in appearance and geometry. Moreover, each subject is asked to perform various dynamic motions, such as expressions and head rotations, which further extend the richness of assets. 3) Rich Annotations: the dataset provides annotations with different granularities: cameras' parameters, background matting, scan, 2D/3D facial landmarks, FLAME fitting, and text description.

Based on the dataset, we build a comprehensive benchmark for head avatar research, with 16 state-of-the-art methods performed on five main tasks: novel view synthesis, novel expression synthesis, hair rendering, hair editing, and talking head generation. Our experiments uncover the strengths and flaws of state-of-the-art methods. RenderMe-360 opens the door for future exploration in modern head avatars. All of the data, code, and models will be publicly available at `https://renderme-360.github.io/`.

# 1 Introduction

Digitalizing human replicas is a perennial topic in both research and commercial communities. It serves as the foundation of many advanced applications, *e.g.,* VR/AR, gaming, and metaverse. Among various tasks, human head avatar synthesis plays a crucial but difficult role. This is because the human head performs significant social functions with appearance, expression, speech, *etc.*, in which even subtle differences between synthesized and real ones can be easily perceived by human eyes to trigger the uncanny valley effect. How to render, reconstruct, and animate a human head with realism reminds a great challenge. Over decades, although numerous approaches have emerged and pushed forward the frontier of *facial* reconstruction [43, 44] and animation [13, 38], general full-*head* level avatar synthesis [56, 3, 53] has only started to actively advance in recent years. Research efforts along human head avatar usually follow the flourishing of deep learning and neural rendering. Such formalizations require large-scale or dense multi-view training datasets to drive progress.

Unlike the efforts on 2D datasets [28, 20], which could utilize Internet-scale data to enhance the quantity and diversity, the path to constructing a 3D/4D repository is difficult. Thus, current human head-related datasets [51–54, 11, 10] have significant limitations on *dataset scale*, *sample diversity*, *photorealism*, *sensory modality*, and *annotation granularity*. For example, Multiface dataset [51] contains facial data with only 13 publicly available subjects, VOCASET [11] focuses on auditory modality while ignoring other facial functions, and HUMBI Face [54] suffers from low resolution with 2 million pixels. The details of the existing head-related datasets' limitations are shown in Table 1. These datasets are valuable. However, they can only enable researchers to study a small set of problems. The progress of human head avatar algorithms also indicates the saturated performances on existing datasets, while the performance gap between standard datasets and real-world scenarios still remains. Moreover, human head avatar synthesis is a complex combination of many fundamental tasks (such as face/head reconstruction, expression animation, and hair modeling/animation), which requires a comprehensive digital asset library to support the exploration. In a nutshell, compared with 2D counterparts, the construction of 3D/4D human head repositories is impoverished.

In this paper, we present **RenderMe-360**, a new publicly available large-scale 4D digital asset library with over 243 million frames that features a wide range of downstream tasks, to boost the development of human head avatar creation. RenderMe-360 goes beyond previous datasets in several key aspects: 1) *High Fidelity:* we set up a high-end data collection system to capture high-resolution raw data of RenderMe-360. With the system, all data is ensured to be captured by 60 cameras at $2448 \times 2048$ resolution and 30 FPS. 2) *High Diversity:* We collect 500 different participants, who come from various countries with diverse ages and cultural backgrounds (illustrated in Figure 3). Specifically, about $25\%$ of them are with designed makeup styles and wearing special decorations, such as ancient Chinese makeup styles with delicate hair accessories. These nature differences of the participants provide ample variety in both appearance and accent. For each subject, we capture 12 expressions, 26 or 42 bilingual speeches, and 12 hairstyles (if not specifically required). These collection protocols further enrich the diversity of motion, modality, and appearance. 3) *Rich Annotations:* We provide

Table 1: **Multi-view head dataset comparison. N**: Dataset is not released, **S**: Scanner, **M**: Mesh, **AU**: Action Unit, **PF**: Per-Frame. Outfit: A variety of clothes-related & accessory-related designs. Motion: head or body motion, but facial changes are not included. Better zoom in for details.

| Dataset | ID | Age | Expression | Sentence | Language | Frame | Era | Ethnicity | Outfit | Accessory | HairStyle | Makeup | Motion | Camera View | Resolution | FPS | Wig Style× | Wig Color | Appearance Annotation | Phoneme-balanced Corpus | PF Face Lmk2d | PF Face Lmk3d | PF Matting | 3DMM-like model | Scan | UV map | Activity Descriptions | PF AU |
|---|---|---|---|---|---|---|---|---|---|---|---|---|---|---|---|---|---|---|---|---|---|---|---|---|---|---|---|---|
| | | | | Diversity | | | | | | | | | | Realism | | | Granularity | | | | | | | | | | | |
| D3DFACS [10] | 10 | - | 19-97AU | ✗ | ✗ | - | ✗ | ✗ | ✗ | ✗ | ✗ | ✗ | ✓ | 6(S) | M | 60 | ✗ | ✗ | ✗ | ✗ | ✗ | ✗ | ✗ | ✗ | ✗ | ✗ | ✗ | ✓ |
| HUMBI Face [54] | 772 | (<10)-(>60) | 20 | ✗ | ✗ | 17.3M | ✗ | ✓ | ✓ | ✗ | ✓ | ✗ | ✓ | 68 | 2MP | 60 | ✗ | ✗ | ✓ | ✗ | ✓ | ✓ | ✗ | ✗ | ✗ | ✗ | ✗ | ✗ |
| Facescape [52] | 847 | 16-70 | 20 | ✗ | ✗ | 16.9K M | ✗ | ✗ | ✗ | ✗ | ✗ | ✗ | ✗ | 68 | 4-12MP | - | ✗ | ✗ | ✗ | ✗ | ✓ | ✗ | ✗ | ✓ | ✓ | ✓ | ✗ | ✗ |
| i3DMM [53] | 64 | 16-69 | 10 | ✗ | ✗ | N | ✗ | ✓ | ✗ | ✗ | ✓ | ✓ | ✗ | 137 | - | - | ✗ | ✗ | ✗ | ✗ | ✗ | ✗ | ✗ | ✓ | ✓ | ✓ | ✗ | ✗ |
| VOCASET [11] | 12 | - | ✗ | 40 | 1 | - | ✗ | ✗ | ✗ | ✗ | ✗ | ✗ | ✗ | 18 | M | 60 | ✗ | ✗ | ✓ | ✗ | ✗ | ✗ | ✗ | ✗ | ✓ | ✗ | ✗ | ✗ |
| Multiface [51] | 13 | - | 65/118 | 50 | 1 | ≈ 15M | ✗ | ✗ | ✗ | ✗ | ✗ | ✗ | ✗ | 40/150 | 3MP | 30 | ✗ | ✗ | ✓ | ✗ | ✓ | ✓ | ✓ | ✗ | ✓ | ✓ | ✗ | ✗ |
| **RenderMe-360** | 500 | 6-88 | 12 | 26/42 | 2 | >243M | ✓ | ✓ | ✓ | ✓ | ✓ | ✓ | ✓ | 60 | 5MP | 30 | 7 × 6 | 127 | ✓ | ✓ | ✓ | ✓ | ✓ | ✓ | ✓ | ✓ | ✓ | ✓ |

multiple types of annotations with different granularities (Table 1), which ensure the compatibility of one single dataset to various tasks and methods. Concretely, we provide annotations in two levels: per-frame annotations, and per-id annotations. The per-frame annotations refer to annotating every frame of the collected data. These per-frame annotations include camera parameters, matting, facial action units, and 2D/3D landmarks. The per-id annotations refer to annotating keyframes for each identity in fine-grained hierarchy, including 3D scans, FLAME fitting, UV maps, and text annotations for appearance and activity descriptions. Our vast exploration space and massive data assets serve as the foundation to investigate the performance boundary of state-of-the-art head avatar algorithms.

Based on RenderMe-360 dataset, we set up benchmarks on five fundamental tasks, *i.e.,* novel view synthesis, novel expression synthesis, hair editing, hair rendering, and talking head generation, with extensive experimental settings evaluated on 16 baseline methods (Table 2). We probe in detail how different factors might introduce the influences to current baseline methods. Our experiments present many new observations and challenges for the research community to catalyze future research on the human head avatar. We hope RenderMe-360 could kickstart research efforts in related areas, and spur new opportunities not only from our formalized benchmarks, but also alternative ones that the community might come up with from our comprehensive, massive, and publicly available dataset.

## 2   Related Works

### 2.1   Multi-View Head Dataset

Data serve as the primary fuel for promoting the development of algorithms. While there are many open-world unstructured 2D datasets [20, 59, 29, 19, 21, 35] or synthetic ones, we focus on those real human heads with structured data. Collecting 3D/4D data is essential for head avatar research in both training and evaluating aspects. In the early days of computer vision, researchers mainly focused on 3D face reconstruction/tracking from data sources that included multi-view cues. In 1999, Blanz and Vetter [3] used a laser scan to capture 3D faces, and proposed to model a morphable model (*i.e.,*3DMM) from the database. As such a piece of equipment is not suitable for dynamic motion tracking, Zhang *et al.* [55] present a multi-camera active capturing system with six video cameras and two active projectors to ensure spacetime stereo capturing. Later on, Paysan *et al.* [37] collect 3D faces by ABW-3D system. D3DFACS [10] introduces a dynamic 3D stereo camera system to capture 4D high-quality scans of 10 performers with Action Unit annotations. Upon D3DFACS, Li *et al.* [22] additionally integrate 4D scans from CAESAR dataset [39] and self-captured ones (from 3dMD system). These datasets are mediocre in texture resolution and quality. Recently, Facescape [52] is proposed to fulfill the raw data quality, in which 3D faces are collected from a dense 68-camera array with 847 subjects performing specific expressions. Whereas, these research efforts are limited to supporting facial shape and expression learning. To take a step further on modeling the entire head, i3DMM [53] dataset is proposed with 64 subjects captured by a multi-view scanning system, called Treedys. Since Treedys is not specifically designed for head-scale capture, the authors apply post-process to crop the head meshes based on 3D landmarks, and removing the rest part of the upper body. HUMBI [54] is a large-scale dataset, which contains different body part collections. As the systems for these two datasets are not customized to best fit head-level capture, they are limited in resolution. Multiface [51] contains head-oriented collections and detailed annotations, but only releases 13 subjects' data. To facilitate multisensory modeling, VOCASET [11] is proposed. It is a 4D speech-driven scan dataset with about 29 minutes of 4D scans and synchronized audio from 12 speakers. Although VOCASET allows training and testing of speech-to-animation geometric models and can generalize to new data, it is limited in the extremely narrow diversity of subjects and onefold task. The other alternative is audio-visual data. These datasets are widely used in audio-visual learning tasks, like lip reading [7, 6], speaker detection [40] and talking head generation [47, 58]. For example, GRID [9] and MEAD [47] are sparse multi-view datasets (four and eight respectively), which are characterized by consistent shooting conditions, carefully designed identity, and corpus distribution. However, the sparsity leaves these datasets more often to be used in 2D methods.

In contrast, our RenderMe-360, is a large-scale multi-view dataset for high-fidelity head avatar creation research. It is under a head-oriented, and high-resolution data capture environment. It contains diverse data samples (with 500 subjects performing various activities, *e.g.,* expressions,

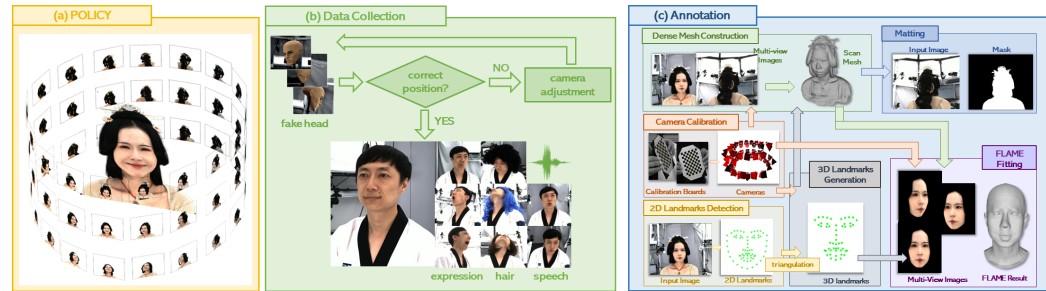

Figure 2: **Overview of data collection pipeline.** (a) Our system, named as POLICY, records subjects' performances in 60 different views. (b) Illustration of data collection process. (c) Illustration of annotation pipeline.

speeches, and hair motions), multi-sensory data, and rich annotations. It is a comprehensive digital asset library that ensures the compatibility of evaluating multiple head avatar tasks in one single dataset. A comparison between RenderMe-360 and other related datasets is shown in Table 1.

## 2.2 Neural Rendering for Head Avatar

Due to space limits, we discuss methods for head rendering in main paper, and unfold the other three directions – hair reconstruction, hair editing, and talking head generation in supplementary.

**Head Rendering.** Head or face priors are often used to condition the neural fields. Such a philosophy can help either improve the robustness or create controllable avatars [16, 2, 5]. Priors like parametric model [4, 22] coefficients, key points, and explicit surface mesh/point clouds are popular ones to be integrated into the framework. For example, NHA [17] presents a framework to learn vertex offsets and attached textures from fitted FLAME surface via coordinate-based MLPs embedded on the surface. NerFACE [15] and IM Avatar [56] use FLAME [22] model expression coefficient to condition the neural field and learn to create an animatable head avatar from monocular video. Taking multi-view images as input condition, Neural Volume [30] models dynamic 3D content with a volumetric representation. MVP [31] replaces the single volume with a mixture of multiple predefined volumetric primitives which improves the resolution and efficiency of volume rendering. To increase the flexibility of re-rendering the avatar in new environments (*e.g.,* novel expression and lighting), PointAvatar [57] presents a paradigm of utilizing point-based representation which achieves fast model convergence by coarse-to-fine optimization. For generalization, KeypointNeRF [32] synthesizes free viewpoints of human heads via multi-view image features and 3D keypoints. In addition, some researchers use cross-domain data such as audio or text to condition the neural fields. For instance, ADNeRF [18] presents a 3D-aware alternative to the 2D talking face pipelines by conditioning the radiance field with both head poses and audio fragments.

# 3 RenderMe-360

In this section, we introduce RenderMe-360 dataset in detail. We start with the description of our capture system (Section 3.1), and move on to an overview process of data collection (Section 3.2). Then, we present the data annotation pipeline (Section 3.3). The whole process is visualized in Figure 2.

## 3.1 Capture System

As illustrated in Figure 2(a), we build a multi-video camera system to record synchronized multi-view videos of human head performance. It contains 60 industrial cameras and covers a field of view of $360°$ left-to-right and over $160°$ up-to-down for video capture at the whole-head level. To ensure encompassing fine details (*e.g.,* hair strands, wrinkles, and freckles), we choose cameras with a high resolution at $2448 \times 2048$. The shutter speed is 30 FPS for capturing fine-grained motion changes. To capture multi-sensory information, a condenser microphone is collocated with the camera system, and under the audio-vision synchronization. Please refer to supplementary for more system details.

## 3.2 Data Collection

The data collection pipeline is illustrated in Figure 2(b). Specifically, to guarantee the valid rate of captured data, we first apply a trial collection with a fake head to check on the operability of equipment and adjustment of camera positions before formal acquisition. After this, we start the formal capture process with the following parts for per-person recording: 1) *Calibration Capture.* We capture camera calibration data before every round of recording. We use a chessboard and move it in front of the cameras at a fixed-order trajectory. 2) *Expression Capture.* We ask each subject to

perform the same expression set, which includes 12 distinctive facial expressions (1 natural and 11 exaggerated expressions) defined in [53]. 3) *Hair Capture.* To cover diverse hair materials and hair motions, we record 12 video sequences (on average) for each *normal* subject, with different hairstyles under three levels –original hair, headgear, and wig captures. Specifically, the collected data includes one motion sequence for the subject's original hair, one for headgear that hides one's hair, and rest sequences for wearing different wigs with random styles and colors. 4) *Speech Capture.* We provide rich corpus that encompasses single words combined sentences, phonetically balanced protocols, and short paragraphs in two languages (Mandarin and English). For each subject, we randomly pick materials from the corpus and ask the subject to speak 26 or 42 phonetically balanced sentences.

We obtain a large-scale dataset of over $800k$ recording videos from $500$ identities at the end, which is gender-balanced, includes multiple ethnicities, and spans ages from 6 to 88 with approximate normal distribution where teenagers and adults form the major part (Figure 3). A more detailed description of our data collection process and related data statistics are discussed in the Supplementary Materials.

### 3.3   Data Annotation

Diverse and multi-granularity head-related annotations are crucial for the research of human head avatar tasks. However, there is still a deficiency of an all-around head dataset with rich annotation in the research community. To facilitate the development of downstream tasks, we provide rich annotations. We also provide a toolbox to automatically label most of the annotations (Figure 2(c)). We unfold the key information in this section. For more details, please refer to Supplementary.

**Camera Parameters.** We estimate extrinsic matrix and rectify intrinsic matrix for each camera via a fine checkerboard pipeline [8]. The process includes checkerboard detection, intrinsic calibration, and extrinsic calibration with multi-view bundle adjustment. To ensure quality, we additionally applied fast novel view synthesis [34], and facial landmark reprojection on multi-view single-frame to eliminate unqualified estimation on camera parameters.

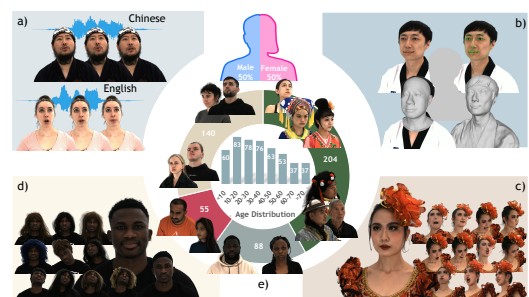

Figure 3: **Key data statistics.** a) 26 or 42 speeches are recorded per subject. b) Annotation example. c) 12 expressions are captured for each subject. d) 10 wigs (on average) are randomly sampled for each normal subject. Original hair and headgear are also captured. e) Distribution of gender, age, and ethnicity. Better zoom in for details.

**2D & 3D Facial Landmarks.** 2D landmarks are per-frame detected via an enhanced version of [50] on selected frontal views, which range from $60°$ left to $60°$ right. With 2D landmarks from multiple views, RANSAC [14] triangulation is applied to obtain 3D landmarks. To guarantee accuracy, low-quality 2D ones are filtered out with spatial and temporal constraints, and 3D results with large re-projection errors are also filtered out. For the frames that are neither precisely calculated on 2D landmarks nor 3D landmarks, we manually label the 2D landmarks and re-run the triangulation.

**Dense Mesh Reconstruction.** Traditional MVS algorithms based on feature points extraction and geometric optimization, such as [46], can only generate irregular point clouds, and have low-quality results in areas of texture missing, such as black hair and dark skin. Therefore, we additionally apply NeuS [48], which uses neural representation for signed-distance-function and optimizes with surface-based rendering results to do multi-view reconstruction and dense mesh extraction. For video sequences, the first frame is optimized from scratch, then the following frames are fine-tuned on the optimized neural representation to accelerate convergence speed.

**Matting.** Reasonable foreground segmentation for human heads is challenging. Since diverse hairstyles and accessories form the long-tail problem. Thus, we develop a united pipeline that combines video-based matting and scan mesh information to improve matting quality. Specifically, we capture the background prior to each round of recordings, and apply RVM [25] to estimate the rough matting result in the first step. We additionally blend depth-aware mask via Z-buffer during rasterization [26] on the scanned mesh to improve matting quality. With multi-view information, the background ambiguity can be distinguished from other views. We use Gaussian Mixture Model to blend the estimations from two models for each pixel[41]. For extremely hard cases where both steps cannot output satisfactory results, we add human-in-the-loop labeling.

**FLAME Fitting.** Since only keyframes are attached with scan meshes to save processing costs, we use two fitting methods in practice – one is fitted with scan mesh, and the other is not. For frames with

scans, we use 3D landmarks to initialize FLAME parameters and optimize via ICP [1]. Since scan shows accurate facial shape in world space, fitting with it preserves better facial contour. For the ones without scans, personalization is designed. The key is to select neutral frames with corresponding scan geometries, and average across these fitting results to generate a personalized template for each subject. We keep shape parameters constant during fitting the expression sequences which lack scans.

**Text Annotation.** To facilitate multi-modal research on human head avatars, we provide text descriptions for the captured videos at unprecedented granularity. These descriptions cover both static and dynamic attributes from four major aspects: 1) *static facial features* of the subjects, where over 90 attributes at general facial appearance, detailed appearance, and lighting condition levels are described; 2) *static information of non-facial regions*, where the texture, material, and shape attributes of subject's accessories (such as necklace, earrings, and hairpin) and hairstyle are defined; 3) *dynamic facial actions*, fine-grained action units (AUs) descriptions based on the FACs system[12] are given; 4) *dynamic video activity descriptions*, where full-sentence annotations of global action sequence descriptions for each captured video are provided.

# 4    Benchmark

Our RenderMe-360 dataset provides various potentials in new research directions and applications for human avatar creation. Here, we build a comprehensive benchmark upon RenderMe-360 dataset, with 16 representative methods on five vital tasks of human head avatars (summarized in Table 2). These tasks range from static head reconstruction and dynamic synthesis to generation and editing. For each task, we set up several experimental protocols to probe the performance limits of current state-of-the-art methods under different settings. *Due to space limits, we present the key insights of the novel view synthesis task in the main paper, and highlight the best/worst results in lavender/gray colors.* **Please refer to Supplementary Materials for benchmarks of the rest four tasks (novel expression synthesis, hair rendering, hair editing, and talking head generation), more implementation details, experiments, qualitative/quantitative results, and discussions.**

## 4.1    Novel View Synthesis

Here, we present novel view synthesis (NVS) benchmark of both case-specific (*i.e.,*Single-ID NVS in sub-section 4.1.1) and generalizable (sub-section 4.1.2) tracks.

### 4.1.1    Single-ID NVS

This case-specific track refers to the setting of training on a single head with multi-view images, which originates from NeRF [33]'s de facto setting, to evaluate the robustness of *static* multi-view head reconstruction. We study four representative methods with two protocols – 1)*#Protocol-1* for exploring methods' robustness to different appearance or geometry factors. The dataset is split into three categories, *i.e.,* Normal Case, With Deformable Accessory, and With Complex Accessory, according to the complexity of appearance and geometry. We discuss the protocol in the main paper; 2)*#Protocol-2* for probing methods' robustness to different camera number and distributions. We discuss this protocol in Supplementary Materials.

Table 2: **Methods for RenderMe-360 benchmarks. M**: Multi-view images,: **S**: Single-view images, **C**: Camera Calibration, **V**: Neural Volumetric, **S**: Neural SDF, **P**: Point-based Representation, **F**: Feature Space, **S**: Static, **D**: Dynamic, **I**: Images Conditioning, **L**: Latent Codes, **P**: Parametric Models **K**: Face Keypoints, **R**: Radiance Field-based, **F**: SDF-based, **N**: Convolution-based, **S**: Case specific, **G**: Generalizable. Better zoom in.

| Attribute | Instant-NGP [34] | NeuS [48] | NV [30] | MVP [31] | IBRNet [49] | KeypointNeRF [32] | VisionNeRF [24] | NeRFace [15] | IM Avatar [56] | PointAvatar [57] | NSFF [23] | NR-NeRF [45] | HairCLIP [42] | StyleCLIP [36] | ADNeRF [18] | SSPNeRF [27] |
|---|---|---|---|---|---|---|---|---|---|---|---|---|---|---|---|---|
| | Novel View Synthesis | | | | | | | Novel Expression Synthesis | | | Hair Rendering | | Hair Editing | | Talking Head | |
| Required Data | M+C | M+C | M+C | M+C | M+C | M+C | M+C | S+C | S+C | S+C | M+C | M+C | S | S | S+C | S+C |
| Representation | V | S | V | V | V | V | V | V | S | P | V | V | F | F | V | V |
| Static/Dynamic | S | S | D | D | S | S | S | D | D | D | D | D | S | S | D | D |
| Conditioning | I | I | I | I | I | I | I | L | L | L | I | I | L | L | L | L |
| Face Priors | ✗ | ✗ | ✗ | P | ✗ | K | ✗ | P | P | P | ✗ | ✗ | F | F | P | P |
| 3D Consistency | R | F | F | R | R | R | R | R | R | R | R | R | N | N | R | R |
| Generalizability | S | S | S | S | G | G | G | S | S | S | S | S | G | G | S | S |

**Settings.** We select 20 identities from the three categories to evaluate the methods. For the training-testing split, we uniformly sample 22 views from all 60 views as test views, and use the rest camera views to train each model. A visualization of camera distribution is shown in Supplementary Materials,

noted as $Cam1$. The four methods for comparison are: Instant-NGP [34], NeuS [48], NV [30] and MVP [31]. The first three methods are originally designed for general-purpose case-specific NVS, and the last one is designed for human head avatar reconstruction. We compute PSNR, SSIM, and LPIPS for rendered novel view images against ground-truth. The quantitative results are listed in Table 3, and the qualitative ones are listed in Supplementary Materials.

**Results.** We observed three key phenomena under *#Protocol-1*: 1) All methods tend to drop the performance lengthways along the Table 3, with the level of accessory complexity increasing. This phenomenon reflects the status quo that we do not yet have one strong paradigm for robust case-specific human head multi-view reconstruction. 2) NeuS yields the best performance on average. There are two possible underlying reasons. First, NV and MVP are dynamic methods while not emphasizing temporal-consistency constraints. Thus, when comparing these methods with static ones under static measurement, the perturbation of data sequences would affect these two methods' construction on dynamic fields to certain degrees. Second, by associating the quantitative results with the qualitative ones, we can find that NeuS performs well in global shape reconstruction with almost no surrounding noise due to its surface representation property, but has a much smoothing surface appearance. In contrast, Instant-NGP and MVP can recover better high-frequency details. MVP uses multiple-primitive representation with different networks to render, equipping the model with a larger representative capacity. Whereas, they produce more surrounding noise. Neural Volume renders images mostly with artifacts. We could draw the idea that surface representation helps the novel view reconstruction in a global shape-forming manner. 3) NV suffers from limited grid resolution (although it uses inverse warping to ease the problem) and inaccurate alpha value estimation. Thus, it introduces more artifacts than others, and is strenuous in reconstructing high-frequency details.

Table 3: **Single ID NVS (*#Protocol-1*).** We evaluate methods under three subsets with levels of complexity.

| Split | Metrics | NGP [34] | NeuS [48] | NV [30] | MVP [31] | Split | Metrics | NGP [34] | NeuS [48] | NV [30] | MVP [31] |
|---|---|---|---|---|---|---|---|---|---|---|---|
| **Normal Case** | PSNR↑ | 24.71 | 26.29 | 19.61 | 23.65 | **With Complex Accessory** | PSNR↑ | 20.54 | 22.89 | 16.46 | 21.5 |
| | SSIM↑ | 0.848 | 0.927 | 0.777 | 0.895 | | SSIM↑ | 0.776 | 0.874 | 0.598 | 0.83 |
| | LPIPS↓ | 0.28 | 0.11 | 0.29 | 0.14 | | LPIPS↓ | 0.36 | 0.16 | 0.44 | 0.18 |
| **With Deformable Accessory** | PSNR↑ | 23.06 | 23.53 | 17.83 | 23.93 | **Overall** | PSNR↑ | 23.21 | 24.67 | 18.56 | 23.1 |
| | SSIM↑ | 0.807 | 0.904 | 0.703 | 0.893 | | SSIM↑ | 0.819 | 0.906 | 0.723 | 0.876 |
| | LPIPS↓ | 0.31 | 0.13 | 0.34 | 0.12 | | LPIPS↓ | 0.31 | 0.13 | 0.33 | 0.15 |

### 4.1.2 Generalizable NVS

This track refers to the setting of training across multiple human heads, and testing on unseen[1] human heads (*i.e.,* new identities) or unseen motions (*e.g.,* expressions) with *conditioning on one or few input images* (as source views). It allows us to evaluate the network's effectiveness in learning priors, and the ability to adapt priors. We investigate three methods under two protocols in the main paper – 1) *#Protocol-1* for investigating methods' generalization ability on geometry deformation via evaluating the generalization ability to unseen expressions on seen identities. 2) *#Protocol-2* for probing methods' capability in learning category-level human head priors via evaluating the generalization ability to unseen identities. We also name this protocol as *Unseen ID NVS*; This setting is challenging as it requires the model to generalize to both new appearances and geometries. To further reveal the factors that might have influences on generalization, we enrich both protocols with four sets of training-testing view settings and three data subsets under different complexity.

**Settings.** We study three generalizable methods: IBRNet [49], VisionNeRF [24], and Keypoint-NeRF [32]. For training-testing identity split, we select a subset from RenderMe-360, with 160 identities for training and 20 for serving as unseen identities. The selected identities are evenly sampled from the three data subsets. We select 7 out of 60 camera views as novel views. Note that, we calculate the metrics in *#Protocol-2* on all expressions. As a consequence, 10 expression structures attached with trained identities are covered in training set, and the rest 2 expression structures are unseen. Such an evaluation strategy provides the feasibility for researchers to analyze their methods' generalization ability on appearance and geometry in both entangled and disentangled aspects.

**Results.** The quantitative results are shown in Table 4. From per method perspective, we draw the consistent conclusions that: 1) random view training could help enhance the model's robustness on both unseen expression and identity tasks; 2) the performance declines in terms of most metrics with the complexity of human head's appearance/geometry increase; 3) the Unseen ID NVS task introduces larger performance drop rate than Unseen Expression NVS. These two phenomena suggest that these generalizable methods could learn priors like the information of 'minimal-accessory'

---

[1]Note that the adjective 'seen' refers to the sample that is used in training, and 'unseen' means the sample is not used as training data.

Table 4: **Benchmark results on generalizable NVS.** The results are evaluated on: (1) unseen expressions of sampled training identities, and (2) unseen identities in each test split. (LPIPS* denotes LPIPS × 1000)

| | Training Setting | Testing Setting | Methods | Normal Case | | | With Deformable Accessories | | | With Complex Accessories | | | Overall | | |
|---|---|---|---|---|---|---|---|---|---|---|---|---|---|---|---|
| | | | | PSNR↑ | SSIM↑ | LPIPS↓* | PSNR↑ | SSIM↑ | LPIPS↓* | PSNR↑ | SSIM↑ | LPIPS↓* | PSNR↑ | SSIM↑ | LPIPS↓* |
| Unseen Expression NVS | Fixed Views | Fixed Views | IBRNet [49] | 23.36 | 0.918 | 144.17 | 20.82 | 0.849 | 197.85 | 20.33 | 0.827 | 187.57 | 21.97 | 0.878 | 168.44 |
| | | | VisionNeRF [24] | 23.57 | 0.905 | 139.52 | 20.42 | 0.846 | 186.04 | 20.89 | 0.835 | 189.60 | 22.11 | 0.873 | 163.67 |
| | | | KeypointNeRF [32] | 19.59 | 0.898 | 127.06 | 17.42 | 0.805 | 213.43 | 16.54 | 0.760 | 205.82 | 18.29 | 0.840 | 168.34 |
| | | Random Views | IBRNet [49] | 24.34 | 0.924 | 140.21 | 20.81 | 0.85 | 189.57 | 20.45 | 0.832 | 179.57 | 22.485 | 0.883 | 162.39 |
| | | | VisionNeRF [24] | 25.79 | 0.914 | 148.70 | 21.43 | 0.883 | 148.90 | 20.37 | 0.87 | 159.50 | 23.345 | 0.895 | 151.45 |
| | | | KeypointNeRF [32] | 16.96 | 0.871 | 170.66 | 16.07 | 0.775 | 256.83 | 14.64 | 0.714 | 270.33 | 16.16 | 0.808 | 217.12 |
| | Random Views | Fixed Views | IBRNet [49] | 23.37 | 0.918 | 144.21 | 20.82 | 0.8487 | 197.85 | 19.79 | 0.803 | 182.85 | 21.84 | 0.872 | 167.28 |
| | | | VisionNeRF [24] | 23.05 | 0.905 | 135.20 | 21.42 | 0.864 | 167.00 | 20.28 | 0.835 | 165.01 | 21.95 | 0.877 | 150.60 |
| | | | KeypointNeRF [32] | 19.74 | 0.902 | 113.5 | 18.05 | 0.817 | 183.66 | 17.02 | 0.778 | 182.08 | 18.64 | 0.850 | 148.19 |
| | | Random Views | IBRNet [49] | 24.38 | 0.924 | 139.71 | 21.02 | 0.850 | 190 | 20.91 | 0.837 | 175.14 | 22.67 | 0.884 | 161.14 |
| | | | VisionNeRF [24] | 28.08 | 0.943 | 97.32 | 23.86 | 0.882 | 150.6 | 23.08 | 0.873 | 133.2 | 25.78 | 0.910 | 119.61 |
| | | | KeypointNeRF [32] | 18.65 | 0.897 | 124.33 | 17.60 | 0.813 | 192.04 | 16.61 | 0.779 | 186.67 | 17.88 | 0.847 | 156.84 |
| | Training Setting | Testing Setting | Methods | Normal Case | | | With Deformable Accessories | | | With Complex Accessories | | | Overall | | |
| | | | | PSNR↑ | SSIM↑ | LPIPS↓* | PSNR↑ | SSIM↑ | LPIPS↓* | PSNR↑ | SSIM↑ | LPIPS↓* | PSNR↑ | SSIM↑ | LPIPS↓* |
| Unseen ID NVS | Fixed Views | Fixed Views | IBRNet [49] | 22.25 | 0.895 | 157.96 | 18.42 | 0.824 | 213.55 | 17.97 | 0.744 | 255.98 | 20.22 | 0.840 | 196.36 |
| | | | VisionNeRF [24] | 21.01 | 0.866 | 146.44 | 18.00 | 0.801 | 216.22 | 17.35 | 0.734 | 262.60 | 19.34 | 0.817 | 192.93 |
| | | | KeypointNeRF [32] | 18.85 | 0.866 | 148.13 | 15.93 | 0.789 | 205.04 | 16.14 | 0.734 | 231.89 | 17.44 | 0.814 | 183.30 |
| | | Random Views | IBRNet [49] | 22.54 | 0.897 | 154.06 | 18.72 | 0.831 | 198.75 | 18.12 | 0.751 | 249.35 | 20.48 | 0.844 | 189.06 |
| | | | VisionNeRF [24] | 24.01 | 0.818 | 150.04 | 18.15 | 0.857 | 198.46 | 19.33 | 0.796 | 197.42 | 21.38 | 0.822 | 173.99 |
| | | | KeypointNeRF [32] | 17.03 | 0.841 | 187.19 | 14.79 | 0.76 | 244.94 | 15.46 | 0.715 | 273.00 | 16.08 | 0.789 | 223.08 |
| | Random Views | Fixed Views | IBRNet [49] | 22.24 | 0.895 | 157.95 | 18.42 | 0.824 | 213.55 | 18.01 | 0.746 | 256.81 | 20.23 | 0.840 | 196.57 |
| | | | VisionNeRF [24] | 21.92 | 0.889 | 139.90 | 18.43 | 0.833 | 176.12 | 18.35 | 0.773 | 223.04 | 20.16 | 0.846 | 169.74 |
| | | | KeypointNeRF [32] | 18.96 | 0.868 | 138.21 | 16.15 | 0.800 | 185.43 | 16.12 | 0.744 | 230.09 | 17.55 | 0.820 | 172.99 |
| | | Random Views | IBRNet [49] | 22.53 | 0.897 | 154.05 | 18.75 | 0.830 | 195.12 | 18.10 | 0.749 | 250.72 | 20.48 | 0.843 | 188.49 |
| | | | VisionNeRF [24] | 24.77 | 0.918 | 110.4 | 20.22 | 0.858 | 149.30 | 19.35 | 0.797 | 196.90 | 22.28 | 0.873 | 141.75 |
| | | | KeypointNeRF [32] | 18.02 | 0.865 | 145.30 | 15.75 | 0.794 | 194.16 | 16.15 | 0.747 | 227.49 | 16.99 | 0.818 | 178.06 |

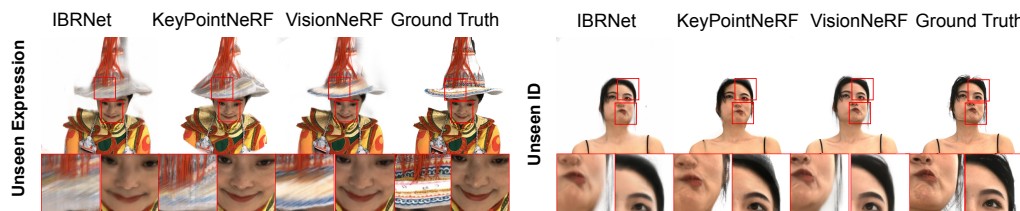

Figure 4: **Qualitative Results of Generalizable NVS (*#Protocol-1&2*).** We illustrate three generalizable methods in two different settings. The regions in red boxes are zoomed in for better visualization.

mean head, and local geometry transformation on a certain level, while still struggling with more diverse scenarios that are long-tail distributed (please associate statistical details in Supplementary Materials). In addition, there are several interesting observations when comparing the three methods: 1) VisionNeRF[24] achieves the best results on average. The robustness might come from its large capacity of learnable variables from a transformer-based structure on image features and the multi-resolution based encoder. 2) IBRNet[49] results in blurry synthesis even under the train and test settings on fixed views. 3) KeypointNeRF [32] falls behind for most of the scenarios, but is in the lead on LPIPS on average. In other words, KeypointNeRF benefits in perceptual measurement like LPIPS while suffering from pixel-wise measurements. We infer the possible reason behind the contradictory metric performances is that – the modules driven by triangulated keypoints provide better feature and view alignments in an explicit manner to help reconstruct the radiance fields. Whereas, such a key insight is a double-edged sword for full human head tasks. Since only the facial region could be well guaranteed with accessible facial landmarks. As a consequence, non-facial regions, like the hat in Figure 4, are more blurry than the facial region and distorted in the geometry aspect. Moreover, KeypointNeRF only renders the intersected frustum regions from source views in practice, which aggravates the performance problem from the full-head measurements. The results turn better when we only calculate regions that KeypointNeRF could render, as shown in Supplementary Materials.

## 4.2 Intra-Dataset Evaluation

**Settings.** To demonstrate the strengths of our dataset's diversity and scale, we first conduct an intra-dataset evaluation with different subset settings. Concretely, we separate the training data into three subsets according to accessories difficulties, namely, 'Subset 1-3' which corresponds to 'Normal Case', 'With Deformable Accessories', and 'With Complex Accessories' respectively. We also randomly sample subsets with 30% and 50% of the training data to investigate the effectiveness of the data scale. We train models ( three generalizable methods [24, 49, 32]) on different subsets, and follow the strategy of *#Protocol-2* in Sec. 4.1.2. To ensure fairness, we stop model training on the same global step with the same learning rate.

**Results.** We span the results of models trained on different training sets by test splits. The quantitative results of IBRNet [49]are visualized in Fig. 5. Please refer to supplementary for detailed numbers and qualitative results. Generally, models trained on one subset achieve relatively better performance on test split of the same category than the ones train on other subsets. For instance, the

model trained on 'Subset 1' gets the best performance in "Normal Case" splits while performing unsatisfactorily in other splits with accessories. With the full training data set, the model tends to overfit less on specific splits and achieves better overall performance. For models trained on smaller random datasets with even distributed data coverage, there is no obvious overfitting on any certain split. While we can observe that with a larger scale of data, the model generalizes better on novel identities. To sum up, we believe the proposed dataset could improve the robustness of generalization methods with both the large scale and the wide range of data distribution.

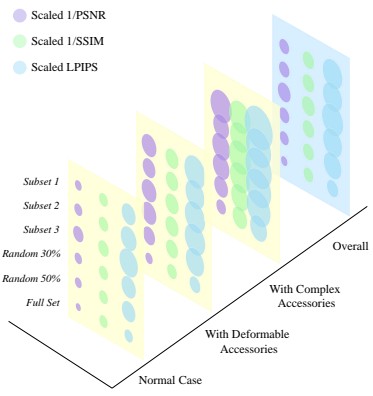

Figure 5: **Intra-Dataset evaluation.** We visualize quantitative results by training and testing on different subsets of the proposed dataset. Smaller bubbles indicate better performances. Better zoom in for details.

### 4.3 Cross-Dataset Evaluation

**Settings.** We conducted cross-dataset experiments to validate the transferability of our dataset. Specifically, we use Multiface dataset [51] for comparison. Similar to intra-dataset evaluation, three aforementioned generalization methods [49, 24, 32] used in Section 4.1.2, are utilized to validate the cross-domain performance of our proposed RenderMe-360 and Multiface. We rigidly transform the Multiface's camera system to align with our dataset so as to avoid coordinate offsets. For fairness, we also crop and resize both source and target images with $512 \times 512$, and train models with the same global step and learning rate. We also follow the strategy of *#Protocol-2* in Sec. 4.1.2, models trained on both datasets are directly evaluated on test sets without any fine-tuning.

**Results.** We visualize the cross-dataset performance of three methods in Fig. 6, where we can observe that our dataset achieves excellent cross-domain performance in all three generalizable methods [32, 49, 24]. Compared to RenderMe-360, Multiface dataset only contains 13 identities. Thanks to the large data coverage and scale of RenderMe-360, it enables robust generalization on novel identities in both in-domain and cross-domain settings. Especially, when testing on the Multiface test set, the cross-domain models trained on our dataset even outperform the in-domain evaluation of Multiface-trained models.

## 5 Boarder Impact and Limitations

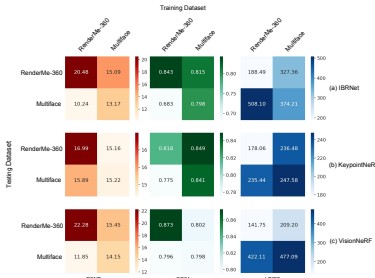

Figure 6: **Cross-Dataset evaluation.** We visualize the quantitative results by training and testing on the different datasets. Better zoom in for details.

Previously, the unavailability of publicly accessible datasets comprising large-scale high-fidelity human heads had impeded advancements in related fields for years. The proposed RenderMe-360 dataset, together with the comprehensive benchmark, is expected to advance the research community in developing high-fidelity head avatars. It also brings new opportunities to many downstream applications in various fields, such as video telephony and the film industry, which shall be beneficial for society. However, there are still several challenges beyond the current scope of this project, that are essential to tackle in the future. We discuss the challenges from both ethical consideration and dataset limitation aspects in the following subsections.

**Ethical Consideration.** Both our dataset and future research efforts based on it, might be applied for malicious purposes like identity theft and fake news. For example, one might use the data to generate fake videos performing various talking interactions with realistic appearances, head/facial/lip motions, and audio. Then, the generated videos might be used in illegal activities such as financial fraud and emotional harm. This might lead to social instability and inflict emotional harm. To alleviate the negative impacts, there are three doable solutions. (1) DeepFake detection algorithms could be applied to evaluate the realism of videos. In this way, the public could be informed whether the video content is real or fake. Besides, our dataset, as well as the data synthesized by algorithms (from our benchmarks), could also serve as references for advancing the development of deepfake detection algorithms. (2) Watermarks could be inserted into rendered images/videos to deter data abuse in practical applications. (3) We caution discretion on behalf of the user, and strongly call for responsible usage of the dataset. For the public good, RenderMe-360 will be open to anyone who

has learned and agrees with our dataset usage agreement, where the data shall be used for research purposes only. Moreover, as our dataset centers on real human data, it is important to consider private information protection and cluster bias problems during the data collection. We summarise the key efforts we made for these aspects as follows – **1) Privacy Authorization and Protection:** All subjects have gone through and signed informed consent before data collection. The consent is based on and compliant with the Personal Information Protection Law of the People's Republic of China (PIPL) standards. The users should learn that the dataset is for research purposes only and prohibits harmful or illegal use. Besides, to protect sensitive personal information, we only collect the necessary information of each subject. We anonymize the data by replacing real names with virtual serial numbers. **2) Fairness and Bias:** We consider demographic representativeness during data collection and minimize personal bias during annotation. Multiple professional annotators label the same data, and the most accurate result is selected through voting. By implementing these measures and promoting responsible usage, we aim to maximize the positive impact of our dataset while minimizing the potential negative consequences.

**Limitations.** RenderMe-360 contains over 243 million video frames with high-fidelity captures and corresponding rich annotations. However, there are some potential limitations specifically affecting RenderMe-360: (1) Adequate data volume towards open-world human avatar generation. Despite the significance of our dataset when compared with existing datasets in terms of data volume, diversity, realism, and granularity, etc, there is still a large magnitude gap between in-lab scale collection and internet scale collection. The multi-view capture setting makes it infeasible to reach the same data volume magnitude with internet scale collection like monocular videos or unstructured 2D images. On the one hand, we will scale up the volume in the future dataset version, with continuous effort to collect more human heads. On the other hand, this limitation could be a starting point for future work on developing algorithms via combining both high-quality multiview data and unstructured open-world collection. (2) Constraints in the data capture system. To achieve high-fidelity capture, we apply uniform lighting, 30fps capture speed, and audio collection with the microphone in one position. Upgrading the system could enable relighting, higher speed capture, subtle movement detection, and multi-positional audio collection *etc*, expanding the possibilities for downstream tasks. (3) There are two potential improvements regarding the data collection setup. Firstly, the naturalness of the wigs in videos may vary due to the random selection process, disregarding the personalized fit for each individual. Insufficiently fitted wigs can lead to a less authentic appearance. To address this, our future endeavors will prioritize the establishment of a more tailored wig collection pipeline, ensuring heightened authenticity. Secondly, an opportunity for improvement lies in expanding our data collection to encompass the depiction of individuals adorned with diverse makeup styles and embellishments. This expansion would augment dataset diversity, bolstering applications in head avatar generation and editing. (4) From a benchmark construction perspective, it is difficult to thoroughly investigate all aspects of the dataset and evaluate all state-of-the-art methods in a single publication. Thus, we regard the benchmark construction upon RenderMe-360 as an ongoing mission. We also welcome contributions from the research community to foster development in this area together.

# 6 Conclusion

We build a large-scale 4D human head dataset and relative benchmarks, RenderMe-360, for boosting the research on human head avatar creation. Our dataset covers 500 subjects with diverse appearances, behaviors, and accents. We capture each subject with high-fidelity appearance, dynamic expressions, multiple hairstyles, and various speeches. Furthermore, we provide rich and accurate annotations, which encompass camera parameters, matting, 2D/3D facial landmarks, scans, FLAME fitting, and text descriptions. Upon the dataset, we conduct extensive experiments on the state-of-the-art methods to form a comprehensive benchmark study. The experimental results demonstrate that RenderMe-360 could facilitate downstream tasks, such as novel view synthesis, novel expression synthesis, hair editing, and talking head generation. We hope our dataset can unfold new challenges and provide the cues for future directions of related research fields.

## Acknowledgment

RenderMe-360 is constructed under OpenXDLab – an open platform for X-Dimension high-quality data. This study is supported under the RIE2020 Industry Alignment Fund Industry Collaboration Projects (IAF-ICP) Funding Initiative, as well as cash and in-kind contribution from the industry partner(s). It is also partially supported by Singapore MOE AcRF Tier 2 (MOE-T2EP20221-0011, MOE-T2EP20221-0012), NTU NAP.

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
