# OpenReview forum: "RenderMe-360: A Large Digital Asset Library and Benchmarks Towards High-fidelity Head Avatars"
_NeurIPS.cc/2023/Track/Datasets_and_Benchmarks — NeurIPS 2023 Datasets and Benchmarks Poster_

### Official Review · Reviewer_q6aZ · 2023-07-21
**The RenderMe-360 dataset proposed in the paper holds much potential for the progress of head avatar generation**

**Rating:** 7
**Confidence:** 5
**Clarity:** The paper is overall clear and easy t…

**Strengths:**

The authors provide detailed data capture and processing pipeline for reference and the dataset holds enough diversity. Moreover, they present a comprehensive benchmark using the collected data with 16 methods, showing the potential of the dataset on various tasks of human head avatars. As the authors promise to make all the data, code, models, and annotations public, I believe that the dataset can facilitate the development of the community.

**Additional Feedback:**

Please refer to the ‘Opportunities for improvement’ and ‘Correctness’ part.

**Correctness:**

The paper is well-organized with a detailed demonstration of the data capture, processing, and possible benchmarks. The experiments seem sound and appropriate.

**Documentation:**

There is sufficient detail on data collection and processing in the paper, but lacks enough discussion on the ethical and responsible use.

**Ethics:**

The dataset contains a large number of face portraits of subjects, which is a matter of personal privacy, but I have seen very little information about privacy authorization in the paper. I hope the author could explain this issue and supplement related content to the manuscript.

**Limitations:**

I suggest the authors add more discussion on the ethical consideration of the dataset in the main paper.

**Opportunities For Improvement:**

The wig shown in the video is not natural sometimes. I also suggest the authors consider the possibility of collecting data like the same human with different makeup styles and decorations.

**Relation To Prior Work:**

The paper has made comparisons with the previous dataset to show the superiority of the RendeMe-360 on diversity, realism, and granularity.

**Summary And Contributions:**

The paper presents a large-scale 4D human head dataset named RenderMe-360 which contains 500 identities with over 800k video sequences recorded by 60 cameras at 30fps. The paper also provides annotations for the data with per-frame and per-id ones. Results show that these data can be used in many head avatar-related areas and brings more possibility to future research.

---

> ### Author Response · Authors · 2023-08-12
> **Review response**
>
> We sincerely thank the reviewer for your constructive comments and recognition of this work. Below, we respond to each question raised by the reviewer.
> > **Q1**. The wig shown in the video is not natural sometimes.
>
> **A1:** The main reasons for some cases that might not look very natural when wearing wigs are two aspects.  (1) The wigs for each actor to wear are randomly selected from our wig repertory,  while neglecting the customized fit of those wigs for each person. Since different people vary in head shape and amount of hair volume, an unfit wig will lead to a less natural look. The underlying reason for random wig selection is that our primary focus on wig-wearing collection is to help the research on static/dynamic hair rendering. Given hair rendering poses great challenges due to hair's intricate micro-scale structure, voluminous nature, complex motion, and self-occlusion, we place emphasis on collecting a wide range of hair structures, reflection effects through diversifying hair color, and large hair motion.  (2) Although we've tried our best to prepare high-quality wigs, it is still possible to notice a disparity in appearance between the wigs and natural human hair, particularly in the case of exaggerated hairstyles.
>
> We consider adding a more customized wig collection pipeline in the future to better ensure authenticity, by inviting a professional company to customize more authentic wigs and select wigs that customized fit to each performer.
>
> > **Q2**. The authors could consider the possibility of collecting data like the same human with different makeup styles and decorations.
>
> **A2:** We thank the reviewer for this very meaningful suggestion. Collecting the data of the same people while with different makeup styles and decorations can help increase the data diversity, and serve more downstream applications in the field of head avatar generation and editing, like makeup generation, makeup styles transfer, decoration generation, etc. Currently, we plan to continue data collection by considering the following parts:
>
> (1) Actors with special make-up to mimic old people. We already started the trial collection of this part these days.
>
> (2) Each actor has one make-up-free look and multiple make-up looks from different era/style themes, and with wearing decorations of different materials/shapes.
>
> > **Q3**. Add more discussion on the ethical consideration of the dataset in the main paper.
>
> **A3:** We agree that extensive discussion of ethical considerations is needed for datasets involving real human data, especially sensitive attributes related to personal information. Due to the page limitation, we did not discuss ethical considerations in the manuscript. According to your suggestion, we have added a new section in the revised version to discuss ethical and responsible use. Here, we summarise the key points as follows.
>
> (1) Privacy Authorization and Protection. (i) All subjects have gone through and signed the informed consent before data collection. The consent is based on and compliant with the Personal Information Protection Law of the People's Republic of China (PIPL) standards.  Besides, the consent clearly outlines the terms of use for the dataset,  with explicit notes the usage/exhibition of the data should be for research purposes only. We will enforce the rights of the consent in the distribution of our dataset. (ii) To protect sensitive personal information, we only collect the necessary information of each subject, and remove personal identification by replacing real name with virtual serial number.
>
> (2) Fairness and Bias. (i) For data collection, we consider demographic representativeness in age, gender, race, etc. (ii) For data annotation, we try our best to avoid personal bias.  Concretely, as text description might be affected by subjective feelings, we require five professional annotators to label the same data source first, and then vote on the five results to select the most precise one.
>
> &nbsp;
>
> We respond to ethical concerns in more detail in the available ethics reviews.  Please don’t hesitate to let us know if there are any additional clarifications or more details that we can offer!

---

> > ### Comment · Reviewer_q6aZ · 2023-08-30
> >
> > Thank the authors for the response, which partially solved my concerns. So I raised the score to Good paper, accept.

---

### Official Review · Reviewer_8MTN · 2023-07-22
**RenderMe-360: A Large Digital Asset Library and Benchmarks Towards High-fidelity Head Avatars**

**Rating:** 9
**Confidence:** 3

**Strengths:**

The proposed dataset features a number of strengths in comparison with the state-of-the-art datasets:
1) large diversity: 500 subjects, different ages, ethnicities, cultures
2) full coverage and high quality: subjects are captured under 360 degrees using 60 synchronized 2K cameras
3) rich annotations, including cameras' parameters, 2D/3D facial landmarks, and text description 90 attributes
4) design of a comprehensive benchmark based on RenderMe-360 dataset, including novel view synthesis, novel expression synthesis, etc, and evaluation of several state-of-the-art algorithms for each benchmark task

**Additional Feedback:**

a significant dataset

**Clarity:**

the writing of the paper needs to be improved, in terms of expressions and style. I suggest that the authors invite a native English proofreader to review the paper.

**Correctness:**

yes, the dataset seems to be constructed in a sound way; the proposed benchmark makes sense as well

**Documentation:**

Yes, the supplemental material answers all the questions

**Ethics:**

this is not discussed in the paper

**Limitations:**

potential negative societal impact is not discussed ; the limitations of the dataset is not discussed either.

**Opportunities For Improvement:**

The writing need to be improved and proofread if possible by a native English people

**Relation To Prior Work:**

well analyzed, in particular through a simple synthesized table

**Summary And Contributions:**

The paper proposes a novel large-scale 4D human head dataset, namely RenderMe-360, captured from 500 identities using multi-view cameras at 30 FPS.

---

> ### Author Response · Authors · 2023-08-12
> **Review response**
>
> We sincerely thank the reviewer for your insightful comments and recognition of this work. We have polished the paper writing in the revised version according to your suggestion. Below, we provide point-to-point responses to the rest of the questions:
>
> > **Q1**. Potential negative societal impact is not discussed.
>
> **A1:** We construct RenderMe-360 for the purpose of advancing the research community in developing high-fidelity head avatars. It also brings new opportunities to many downstream applications in various fields, such as video telephony and the film industry, which shall be beneficial for society. However, both our dataset and future research efforts based on it, might be applied for malicious purposes like identity theft and fake news. For example, one might use the data to generate fake videos performing various talking interactions with realistic appearances,  head/facial/lip motions, and audio. Then, the generated videos might be used in illegal activities such as financial fraud and emotional harm.
>
> To alleviate the negative impacts, there are two doable solutions. (1) DeepFake detection algorithms could be applied to evaluate the realism of videos. In this way,  the public could be informed whether the video content is real or fake. Besides,  our dataset, as well as the data synthesized by algorithms (from our benchmarks), could also serve as references for advancing the development of deepfake detection algorithms. (2) We caution discretion on behalf of the user, and strongly call for responsible usage of the dataset. For the public good, RenderMe-360 will be open to anyone who has learned and agrees with our dataset usage agreement, where the data shall be used for research purposes only.
>
> > **Q2**.  The limitations of the dataset are not discussed.
>
> **A2:**  There are some potential limitations specifically affecting RenderMe-360: (1) Adequate data volume towards open-world human avatar generation. Despite the significance of our dataset when compared with existing datasets in terms of data volume, diversity, realism, and granularity, etc,  there is still a large magnitude gap between in-lab scale collection and internet scale collection. The multi-view capture setting makes it infeasible to reach the same data volume magnitude with internet scale collection like monocular videos or unstructured 2D images. On the one hand, we will scale up the volume in the future dataset version, with continuous effort to collect more human heads. On the other hand, this limitation could be a starting point for future work on developing algorithms via combing both high-quality multiview data and unstructured open-world collection. (2) Data capture system limitation. To achieve high-fidelity capture, we apply uniform lighting,  30fps capture speed, and audio collection with the microphone in one position. We could update our system to enable relighting, higher speed capture of more subtle movements, and multi-positional/directional audio collection. These would help extend the range of downstream tasks. (3) From a benchmark construction perspective, it is difficult to thoroughly investigate all aspects of the dataset and evaluate all state-of-the-art methods in a single publication. Therefore,  we regard the benchmark construction upon RenderMe-360 as an ongoing mission. We also enthusiastically welcome and advocate the contributions from the research community, to foster the development in the field of human head avatars together.
>
> &nbsp;
>
> We have added the above discussions to the revised version.  Please don’t hesitate to let us know if there are any additional clarifications or more details that we can offer!

---

### Official Review · Reviewer_CynN · 2023-07-27
**Good paper**

**Rating:** 7
**Confidence:** 4
**Correctness:** Yes
**Clarity:** Yes

**Strengths:**


Strengths:
The paper is well-written and easy to understand.
The built dataset is very large with high fidelity, high diversity, and rich annotations.
This paper evaluates 16 state-of-the-art methods performed on five main tasks: novel view synthesis, novel expression synthesis, hair rendering, hair editing, and talking head generation.



**Additional Feedback:**

No

**Documentation:**

Yes

**Limitations:**

Weaknesses:
1. The dataset covers 500 different identities. I wonder how many old people and children are included? Additionally, I am curious about how the race is distributed across the dataset.

**Opportunities For Improvement:**

No

**Relation To Prior Work:**

Yes

**Summary And Contributions:**

This paper introduces RenderMe-360, a 4D human head dataset designed to advance head avatar algorithms across different scenarios. It contains massive data assets, with three key attributes: 1) High Fidelity. 2) High Diversity. 3) Rich Annotations. The dataset provides annotations with different granularities, including cameras’ parameters, background matting, scan, 2D/3D facial landmarks, FLAME fitting, and text descriptions. This dataset enables more Avatar including novel view synthesis and etc.

Overall, this is a good paper.

---

> ### Author Response · Authors · 2023-08-12
> **Review response**
>
> We sincerely thank the reviewer for your positive feedback and recognition of this work.  Our responses to your questions are shown below:
>
> > **Q1**.How many old people and children are included in the collected 500 different identities? How is the race distributed across the dataset?
>
> **A1:** We collected  74 old people (equal to or older than 60 years old) and 60 children (under 10 years old).  Our dataset features a wide range of races, with encompassing four ethnicities, i.e., Asian, White, Black, and Brown people. The collected numbers of these four ethnicities are 217, 140, 88, and 55, respectively.  We present these statistics in both Figure 3 (e)(Main Paper) and Figure S15 (a) (Supplemental Materials).
>
> &nbsp;
>
> Please don’t hesitate to let us know if there are any additional clarifications or more details that we can offer!

---

### Author Response · Authors · 2023-08-25
**Reminder: reviewer-author discussion ends on August 29th**

Dear Reviewers Ec5c, WYpK, CynN, 8MTN, and q6aZ,

We'd like to reach out again to check if there were any additional questions or concerns about our rebuttal that we can address before the reviewer-author discussion period ends.

We sincerely thank you again for your great efforts in reviewing this paper, especially for the valuable suggestions that have helped us improve the quality of this work! Please don’t hesitate to let us know if there are any additional details that we can offer!

Paper Authors

---

### Decision · Program_Chairs · 2023-09-22

**Decision:**

Accept (Poster)

**Comment:**

This paper creates a diverse, high-quality dataset for generating human head avatars called RenderMe-360. The details are very rich that can benefit the entire imaging community.  The built dataset is very large with high fidelity, high diversity, and rich annotations. Overall, all reviewers give positive comments, thus the area chair suggests to accept it.